# Stem Cell Therapy for Alzheimer’s Disease: A Scoping Review for 2017–2022

**DOI:** 10.3390/biomedicines11010120

**Published:** 2023-01-03

**Authors:** Yunxiao Duan, Linshuoshuo Lyu, Siyan Zhan

**Affiliations:** 1Department of Biostatistics, Yale School of Public Health, New Haven, CT 06510, USA; 2Department of Environmental Health Sciences, Yale School of Public Health, New Haven, CT 06510, USA; 3Department of Epidemiology and Biostatistics, School of Public Health, Peking University, Beijing 100191, China; 4Research Center of Clinical Epidemiology, Peking University Third Hospital, Beijing 100191, China

**Keywords:** Alzheimer’s disease, stem cell therapy, dementia, scoping review

## Abstract

Alzheimer’s disease (AD) has been a major causal factor for mortality among elders around the world. The treatments for AD, however, are still in the stage of development. Stem cell therapy, compared to drug therapies and many other therapeutic options, has many advantages and is very promising in the future. There are four major types of stem cells used in AD therapy: neural stem cells, mesenchymal stem cells, embryonic stem cells, and induced pluripotent stem cells. All of them have applications in the treatments, either at the (1) cellular level, in an (2) animal model, or at the (3) clinical level. In general, many more types of stem cells were studied on the cellular level and animal model, than the clinical level. We suggest for future studies to increase research on various types of stem cells and include cross-disciplinary research with other diseases. In the future, there could also be improvements in the timeliness of research and individualization for stem cell therapies for AD.

## 1. Introduction

### 1.1. Alzheimer’s Disease and Its Treatments

Among elderly people around the world, Alzheimer’s disease (AD) is one of the major causal factors for mortality [1]. AD is a progressive chronic disease that causes a degenerative disorder of the brain, which gradually worsens memory and eventually leads to disability in performing communication and basic daily tasks [2].

Up until now, numerous approaches to therapeutic drugs are encountering unsatisfying outcomes in improving the cognitive performance of Alzheimer’s disease [3]. The reason why drugs are not curing it well is very complex. One of the reasons is unclear pathology. In other words, the pathogenesis of AD is still under investigation [4]. Another reason is that AD is a chronic disease that requires long-term care. This is unlike treatment for acute diseases where short-term drug therapy is viable, and the outcome is timely. As time progresses, the method of drug intake for AD patients needs careful modification [5]. Most importantly, therapeutic drugs are not able to stimulate the regeneration of neural cells that are already damaged. As the level of severity of AD increases, the lack of vitality of body cells also inhibits the effective transportation of drug molecules [6]. Thus, the efficiency and accuracy of drug therapy are highly restrained.

Besides drugs, most other therapeutic options have not yet shown significant effectiveness in treating AD. Brain-derived neurotrophic factors, despite their importance in maintaining synaptic plasticity in memory and learning, have not yet been discovered in terms of their exact signaling mechanisms [7]. Insulin therapy, despite its potential in safe and short-term symptomatic intervention to delay cognition loss, does not always show significant clinical outcomes in real-life settings [8]. Low-level laser therapy can potentially prevent cognitive impairment by altering brain cell function and neurometabolic pathways, but the optimal wavelength, dosage, and intensity have not been determined for individual treatment. The duration of effects and the chance of chronic repetition are also unclear [9]. Herbal medicine has shown mild effectiveness in treating cognitive deficits of AD, but the action mechanisms of herbs and most of their biochemical formulations are not clearly understood. In addition, since herbal medicines are from natural products, the content and concentration of chemical constituents vary from batch to batch, and could be difficult to standardize [10]. Mitochondrial calcium targeting has the potential to achieve more than symptomatic relief because mitochondrial respiration plays a crucial role in AD etiology [11]. However, why mitochondrial calcium efflux is downregulated early in AD pathogenesis is still a question [12]. Signaling pathways have the potential to provide multilevel, multifaceted, and multitargeted approaches to prevent and treat AD, but it is challenging to implement them in clinical settings due to their complexity [13]. A healthy lifestyle, including moderate physical activity and an appropriate diet, also has an association with lowering the risk of AD. Yet, it is not as effective in the treatment of AD [14].

However, stem cell therapy, which has been developed relatively recently, gives hope for better treatment of AD. Stem cell therapy enhances the level of functional recovery in the central nervous system of the brain [15]. By implementing exogenous stem cells, the depleted neuronal circuitry could be repopulated and regenerated [16]. Stem cell therapy is able to reduce neuroinflammation, which is especially important for patients who develop AD after aneurysmal subarachnoid hemorrhage (SAH) because neuroinflammation plays a vital role in injury expansion and brain damage that eventually cause cognitive decline [17]. Stem cell therapy can also eliminate neurofibrillary tangles and abnormal degradation of proteins, and promote mitochondrial transport to improve cognition [18]. In particular, in the early stages of AD, neural stem cells are able to participate extensively in brain homeostasis, which repairs and exhibits pleiotropic intrinsic properties to mitigate and eventually cure AD [19]. Stem cell-derived exosomes can also have donor-derived properties and minimal immunogenicity, which lower the risk of tumor formation after therapy [20].

### 1.2. Types of Stem Cell for AD Therapy

To thoroughly understand the mechanisms of stem cell therapy for AD, we will evaluate therapies from various types of stem cells and review their development respectively. In general, there are four major types of stem cells used for AD therapy: (1) neural stem cells (2) mesenchymal stem cells (3) embryonic stem cells (4) induced pluripotent stem cells [21].

#### 1.2.1. Neural Stem Cell (NSC)

Up until now, the regeneration of cognitive decline and loss of brain tissue in AD patients have been non-curable. Thus, the majority of effective AD therapies focus on targeting AD pathology in the early stage to preserve cerebrovascular function. Because NSCs contribute considerably to brain homeostasis and repair, they reveal pleiotropic fundamental properties for the treatment of AD in the early stages [19].

In order to develop treatments for AD, it is vital to develop experimental models which represent a specific cellular phase of AD and laborious analysis of the cellular pathological mechanisms [22]. In 2018, McGinley et al. [23] discovered that transplantation of human NSC enhanced cognition of AD in a murine model APP/PS1 (amyloid precursor protein and presenilin 1 mutated mice). The transplantation was targeted to the fimbria fornix, and it significantly improved cognition in the hippocampal-dependent memory tasks at 4 and 16 weeks after transplantation. Furthermore, in 2020, Hayashi et al. [24] modeled human-derived NCS (hNSC) and murine-derived NSC (mNSC) transplantation. Both hNSC and mNSC gave positive results in treating AD.

More recent research has dived into the cellular mechanisms of NSC, and its therapeutic pathology for AD. In 2021, Apodaca et al. [25] discovered that hNSC-derived extracellular vesicles can mitigate the hallmarks of AD. They gave 2-/6-month old 5 × AD mice injections of hNSC-derived extracellular vesicles (EV). NSC treatment significantly decreased dense core amyloid-β plaque accumulation in both age groups, which showed neuroprotective effects for the redress of AD neuropathologies. In 2022, Reveulta et al. [26] studied microglia-mediated inflammation and NSC differentiation in AD, and the possible therapeutic effect of K(V)1.3 channel blockade. They concluded that K(V)1.3 blockers hinder microglia-mediated neurotoxicity in culture, reducing the manifestation and construction of the pro-inflammatory cytokines through NF-kB and p38MAPK pathways.

In general, NSC therapy has developed more advanced and detailed pathology mechanisms, with greater effectiveness in treating AD in the early stage.

#### 1.2.2. Mesenchymal Stem Cell (MSC)

MSC is the most studied type of stem cell in stem cell therapies for AD, due to its excellent accessibility and wide range of differentiating potential. It can be administered intravenously to perform blood–brain barrier penetration with low immune response. In particular, MSC-derived exosomes (MSC-exos) are able to have donor-derived properties with minimal immunogenicity. MSC-exos also have little risk of forming tumors post-therapy, which make them a promising treatment for AD [20].

Several pre-clinical research studies have received significant results in recent years. In 2019, Zaldivar et al. [27] discovered that MSC-exos could increase neural plasticity and enhance cognitive impairment. They injected amyloid-β 1–42 aggregates into the dentate gyrus of murine models bilaterally, and performed novel object recognition tests on days 14 and 28. Results indicated that MSC-exos stimulated neurogenesis in the subventricular zone. In 2020, Nakano et al. [28] discovered that bone marrow-derived MSC (BM-MSC) could enhance cognitive impairment in an AD model by enhancing the expression of microRNA-146a in the hippocampus. BM-MSC were injected intracerebroventricularly into the choroid plexus in the lateral ventricle, and secreted exosomes in the cerebrospinal fluid. In vitro experiments illustrated that exosomal miR-146a from BM-MSC was absorbed in astrocytes, and the level of miR-146a was increased. As the key to forming synapses, astrocytes restore cognitive function and mitigate AD. In the same year, Wei et al. [29] also investigated whether MSC-derived exosomal miR-223 regulates apoptosis of neuronal cells. MSC-derived exosomal miR-223 targeted PTEN, thus activating the PI3K/Akt pathway to inhibit neuron apoptosis, and hence become a potential treatment for AD.

Despite extensive pre-clinical research, research on the clinical level has also made remarkable progress recently. In 2021, Kim et al. [30] performed an intracerebroventricular injection of human umbilical cord blood MSC (hUCB-MSC) in AD patients in a phase I clinical trial. They recruited nine mild-to-moderate AD patients and injected low and high doses of hUCB-MSC, respectively. All adverse events subsided within 36 h, and their symptoms of AD were mitigated.

Overall, MSC therapy reduces neuroinflammation by eliminating amyloid-β, tangles in neuro fibers, and abnormal degradation of proteins. MSC therapy promotes blood–brain barrier and autophagy-related recoveries, regulates acetylcholine levels, and improves cognition of the brain [18].

#### 1.2.3. Embryonic Stem Cell (ESC)

Because there are still ethical and immunogenic limitations to using ESC for treating AD [18], clinical implementation of ESC-based therapy may not be applicable in the short-term. However, there have been a few pre-clinical studies that have shown progress in using ESC to model AD pathology.

On the genetic and cellular level, ESC modeling has made much progress. In 2019, Ubina et al. [31] modeled human ESC on Aβ-dependent neurodegeneration. An allele of APP locus was modified to express Aβ_40_/Aβ_42_ secretory so that the edited allele expression could pass the amyloidogenic APP processing pathway. After neural differentiation, pathway analysis indicated downregulation of the extracellular matrix and over-expression in cilia functions. In 2021, Fan et al. [32] discovered that SIRT1 controls sphingolipid metabolism and neural differentiation of ESC through c-Myc-SMPDL3B. They focused on sphingolipids because they are vital structures of the cell membrane, which regulate cell differentiation and apoptosis. In AD patients, there is a deficit in creating enzymes to remove excess levels of sphingolipids, which eventually leads to neurodegeneration. In particular, SIRT1, an NAD(+)-dependent protein deacetylase, regulates the degradation of sphingolipids by increasing the production of the enzyme SMPDL3B. Therefore, targeting SIRT1 may offer innovative strategies to treat AD. Notably, SIRT1 is sensitive to high-fat diets; therefore, maternal obesity could be a cause of AD as infants develop into adulthood.

In addition, there are also studies with animal models on ESCs. In 2020, Kim et al. [33] investigated the efficacy and feasibility of intra-arterial administration of ESC in an animal model of AD. MSC significantly inhibited Aβ-induced cell apoptosis in the hippocampus, and increased autophagolysosomal clearance of Aβ. MSC-treated mice performed with higher memory ability than those with only Aβ injection.

#### 1.2.4. Induced Pluripotent Stem Cell (iPSC)

iPSC is a technology in which somatic cells are reprogrammed to pluripotent stem cells, creating an optimal physiologically relevant model that maintains the donor’s genetic identity. iPSC can unlimitedly self-renew in vitro, and differentiate into various cell types, which gives hope to model and cure AD [34].

On the genetic and cellular level, there have been multiple studies on iPSC therapy for AD. In 2020, Butler et al. [35] discovered the genetic relevance of human iPSC-derived microglia (iMG) to AD. Microglia are the major immune cell in the brain that imply the pathogenesis of AD. Using gene expression specific to cell type, they showed that iMG cells are genetically relevant to AD. In 2020, Zhang et al. [36] found that human iPSC-derived neural cells from AD patients showed various susceptibilities to oxidative stress. The oxidative stress response of neural cells is a vital mechanism for cognitive dysfunction and aging in AD. Under exposure to H_2_O_2_, the vitality and neurite length of human iPSC-induced neurons reduced significantly. Due to the oxidative property of neuron cells, there is a potential to treat AD by targeting the de-oxidization of the neurons.

In a mouse model of AD, iPSC-derived neural precursors showed improvement in memory and synaptic abnormalities. Researchers injected mouse iPSC-derived neural precursors (iPSC-NPCs) stereotaxically into the hippocampus of mice. Mice with iPSC-NPCs transplantation revealed improvement in synaptic plasticity and reduced AD brain pathology, including decreased tangles deposits and amyloid [37].

iPSC can also be used for drug screening and testing for AD. The flexibility of iPSC includes non-invasive harvesting compatibility and sourcing from patients with AD [38]. In addition, compared to ESC as we discussed in the Section 1.2.3, iPSC is able to reprogram cells without embryo destruction and with negligibly invasive processes [39]. iPSC could create neuronal cells from specific patients, and eradicate the drawback of species-specificity inherent when using animal models. There are also a few novel technologies that can be combined with iPSC models to treat AD, including organoid technology, genome editing, deep learning artificial intelligence, and single-cell RNA sequencing [40].

### 1.3. Overview of Study and Objective

Despite the promising future of stem cell therapy on AD, most research remains in the pre-clinical stage [41]. Therefore, it is worthwhile investigating why there is a gap between animal models and clinical applications, and how to improve it.

This scoping review was conducted to map the research completed in this area systematically, and to identify any existing gaps in knowledge. The following research question was formulated:


*‘How did the different types of stem cells used for Alzheimer’s disease in studies on the cellular level, animal model, and clinical level imply their effectiveness?’*


## 2. Methodology

### 2.1. Search Strategy

This scoping review follows the Preferred Reporting Items for Systematic Reviews and Meta-Analyses (PRISMA) guidelines [42]. The review was performed as described in Figure 1, including papers extracted from PubMed, Web of Science, EMBASE, and Scopus. The papers were intentionally selected to be published in the range of years from 2017 to 2022, i.e., the most recent 5 years, in order to evaluate stem cell therapy’s effectiveness on Alzheimer’s disease using up-to-date data.

The following search query terms were used in the searching databases:

((((Alzheimer[Title/Abstract]) OR (AD[Title/Abstract]) OR (MCI[Title/Abstract]) OR (mild cognitive impairment[Title/Abstract])) AND (stem cell[Title/Abstract]) AND ((therapy[Title/Abstract]) OR (treatment[Title/Abstract]) OR (medication[Title/Abstract]))) NOT (Review)) NOT (Parkinson).

The search was completed in October 2022.

During data selection and review, the search of literature was completed by two researchers in independent locations, and they completed the prepared results. When there was any disagreement between the researchers during the process, they would review the main text together in a research meeting to achieve a consensus result.

### 2.2. Inclusion and Exclusion Criteria

Studies on stem cell therapy for Alzheimer’s disease from 2017 to 2022 were included. Both stem cell therapy and AD had to be mentioned in the article for the studies to be eligible. To be included, the studies also needed to address the therapeutic effects of stem cells and how they relate to the treatment of AD.

After screening and removing duplicated articles, 179 full-text articles were assessed for eligibility, and 117 of them were excluded for the following three reasons:Reason 1 (*n* = 41): “stem cell” was mentioned in the article, not as a means of therapy but as a medium for other medical treatments or preclinical research. For instance, Kim et al. [43] mentioned induced pluripotent stem cells (iPSCs) only as a model for AD to aid their research on estrogen’s mitigating effect on AD. Even though this kind of research was related to stem cells, it did not have a significant relationship with stem cell therapy and thus was excluded from our records.Reason 2 (*n* = 42): the article contained brief mentioning of AD, but was not targeted toward AD. For instance, Rajan et al. [44] mentioned a collection of neurodegenerative diseases, with AD just as a sub-section of the entire article. Even though this article explained well the mechanisms of stem cell therapy, it did not include enough information for the treatment of AD and thus could not be included in our records.Reason 3 (*n* = 34): the exposure group (i.e., treated with stem cell therapy) could not be qualified as eligible to be included in the synthesis. For instance, Campos et al. [45] compared two types of stem cell therapies (neural and mesenchymal), instead of one with a control group. Therefore, it is difficult to qualify and quantify the effects of their research.

Overall, 62 studies were included in the qualitative synthesis. Of the 62 articles, 12 were research on the cellular level, 43 were research on an animal model, and 7 were research on the clinical level.

## 3. Analysis of the Included Studies

In Figure 2, we summarized the distribution (number of studies) for different types of stem cells on (Figure 2a) cellular level, (Figure 2b) animal model, (Figure 2c) clinical level. Clearly, there were many more types of stem cells included in animal model studies than at the cellular level and clinical level. The aggregate number of studies was also greater in animal model studies.

In Figure 3, we collected the effects of different types of stem cells on treating AD, including studies from cellular level, animal model, and clinical level. The goal of all was to mitigate and ultimately cure AD. To achieve this, the research, with different types of stem cells, showed variety in levels of progress. In particular, MSCs showed the most significant effects due to their popularity. The types of stem cells are listed according to their effectiveness from the top (MSC being the most effective) to the bottom (several other types of stem cells).

In the following sections, we are going to present the details of studies on cellular level, animal model, and clinical level, respectively.

### 3.1. Cellular Level

In general, studies on the cellular level of stem cell therapy for AD centered on three types of stem cells: MSC, iPSC, and NSC. Among them, iPSC is the most studied type of stem cell. Most notably, modeling with iPSCs derived from AD patients showed the significance of Apolipoprotein E4 (APOE4) variant as a risk factor for AD. [46,47] MSC is the second most studied type of stem cell among cellular level studies, and there are various sub-types of MSCs in the studies, including: BM-MSC (bone-marrow mesenchymal stem cell), ucMSC (human umbilical cord mesenchymal stem cell), hUCB-MSC (human umbilical cord blood derived mesenchymal stem cell), and hucMSC (human umbilical cord derived mesenchymal stem cell). ucMSC, hUCB-MSC, and hucMSC refer to the same type of stem cell, but we named them differently to show respect for the nomenclature used in different studies. Every stem cell type starts with an “h” referring to a human-oriented stem cell, for instance hNSC is human NSC, and hiPSC is human iPSC. The types of stem cells were also categorized according to their orientation. Among the 12 studies, 1 is a study with animal-induced stem cells, and 11 are studies with human-oriented stem cells. Table 1 is a detailed summary of the control group, intervention, and measured outcomes of interest for studies on the cellular level.

### 3.2. Animal Model

There are many more studies and types of stem cells included in our record of studies for a murine model than on the cellular level. A murine model is the most widely used animal model for studies on stem cell therapy for AD. The benefit of using an animal model is the ability to evaluate the effectiveness of therapies on a tissue level, and this case the brain. A murine model also allows for the conduction of experiments with a larger and more flexible sample size than in clinical research, with minimal concern of ethical issues.

Among various types of stem cells, MSC is the most prevalent type, including many subtypes. sRAGE-MSC is a soluble receptor for an advanced glycation end-products derived mesenchymal stem cell; ES-MSC is ESC derived MSC; AD-MSC is adipose-derived MSC. MFSCE is a membrane-free stem cell, which is a component of adipose-tissue-derived stem cells, where ATSC is an adipose tissue mesenchymal stem cell and hADSC is a human adipose-derived stem cell. In addition to MSCs, there are a variety of other types of stem cells. SCF refers to stem cell factor; hNTSC is human neural crest-derived nasal turbinate stem cell, and SHED is a stem cell from human exfoliated deciduous teeth.

Table 2 is a detailed summary of the control group, intervention, and measured outcomes of interest for the animal model studies.

### 3.3. Clinical Level

In our records, the published results of stem cell therapy for AD on the clinical level, from 2017 to 2022, were much less than the number of those for the animal model. Most clinical-level studies include iPSC because stem cells were generated from specific patients. Table 3 is a detailed summary of the control group, intervention, and measured outcomes of interest for studies on the clinical level.

## 4. Discussion

In our review, we summarized various types of stem cell therapies for AD in the time period from 2017 to 2022. We categorized studies according to whether they were on the cellular level, animal model, or clinical level. We did so to visualize the abundance of research at the pre-clinical level, compared to the deficiency in mature clinical level experiments. The reason behind massive pre-clinical studies is the unclear pathology for AD with stem cell therapy, despite its potential effectiveness. If the pathology is clear and acknowledged, there should be more clinical-level experiments than pre-clinical studies because then studies would move forward to applications in clinical settings. Therefore, the current primary focus of research should be on exploring effective pathologies for potential treatments, and clinical level studies would provide support for the pre-clinical studies.

Among pre-clinical studies, we discovered that MSCs had the maximum quantity of studies, and the most significant effect. Two remarkable effects of MSCs are (1) increase in telomerase activity, and (2) decrease in tau phosphorylation, which restored hippocampal neuronal morphology and improved brain glucose metabolism. MSC is also the only type of stem cell used in clinical level studies that progressed to phase I clinical trials. While iPSC also has applications on the clinical level, it is a reversed approach that generates stem cells from AD patients to aid in analysis based on patient-generated stem cell genetics. Other types of stem cells showed potential for treatments in pre-clinical studies, but not in clinical level studies. Therefore, we suggest future researchers to explore clinical outcomes of MSCs, and expand on the types of other stem cells in the pre-clinical stage. These types of stem cells could include neural crest stem cells, hematopoietic stem cells, and so on [41].

We also encourage cross-disciplinary research with other diseases. Stem cell therapy can not only treat AD, but also many other types of disease such as cancer. In particular, one study has shown that cancer and AD are at opposite sides of the cell division spectrum, and hence cannot happen at the same time [102]. Using this feature, scientists can potentially lower the side effects of stem cell therapy on AD patients who have the risk of developing tumors.

In addition, researchers could create a loop of response from pre-clinical studies and the medical record system in a timely manner, in order to stimulate the efficiency of feedback for therapeutic methods. In other words, there should be instant communication between pre-clinical research and clinical data, without active searching from researchers or physicians. When we used the search engines such as PubMed, they displayed all research in the field according to relevance and time, but did not show a map of correlation between individual experiments. Additionally, since AD is a chronic disease, the time span is relatively long compared to other acute diseases. It would be crucial to track the timing of treatments to find the optimal proportionality of therapeutic injection for each stage of AD. When there are outliers in research or clinical data that are not in-line with therapeutic expectations, secondary-level analysis can be performed on the frequency of outlier occurrence chronologically to track the potential causes, and to contribute to better cure methods and prevention.

Despite the potential effectiveness of stem cell therapy for AD, there are also a few limitations regarding each specific type of stem cell. First, there is a lack of application of NSCs to large-scale clinical trials in AD [103] because isolation and enough expansion of NSCs from the central nervous system are difficult in vitro, and require culture medium supplemented with mitogenic growth factors, such as epidermal growth factor and basic fibroblast growth factor [104]. For iPSCs, there is risk of tumorigenicity and infection, especially in derived donor cells associated with iPSCs-based therapy for AD. These potential side effects bring about a lot of concerns from patients and physicians, especially when the cure for AD is not ascertained [105]. For ESCs, there are also side effects of tumor formation and graft failure, which could negatively impact the well-being of patients post therapy [106]. For MSCs, there is potential organelle dysfunction after therapy, which could be detrimental to the health of the patient [20]. In addition, there are also ethical issues and technical limitations in stem cell therapy. Current ethical concerns for stem cell therapy are centered on the unlimited differentiation potential of iPSCs, which can be used in human cloning [107]. Several technical limitations of stem cell therapy include immunogenicity and limited cell survival in vivo [108].

Individualization is also a key factor that prevents pre-clinical studies from being applied in clinical settings. For cell and animal models, there is less of a concern for the variation of each individual because the animal brain is much less complex than the human brain. For each human individual, however, the brain is much more sophisticated. Physicians also cannot track the memories of their patients because (1) the patients cannot recall everything regarding their previous life experiences, and (2) there is not a medical record of all the brain statuses or activities that can be referred to. A simple CT scan is sufficient for acute brain injuries, but not enough for long-term dementia-like AD. Therefore, the potential solution for this problem is first to ask each patient to evaluate their health situation and involve them in their treatment procedures so that physicians can get first-hand medical data, and second, to make their medical treatment (in this case stem cell therapy) flexible and adjustable as the development of AD grows more or less severe. Physicians and researchers should also consider the patient’s age, sex, family history, social background, and previous health records to make the treatment more suitable for their needs.

Furthermore, there are always concerns about the commercialization of stem cell therapy for AD because it would negatively influence the profits of already-existing drugs. However, these two therapies need not have absolute conflict. In one study, it is shown that iPSC and herbal small-molecule drugs can combine and treat AD more effectively [109]. Furthermore, stem cell therapy is currently not affordable for many patients with lower income, and it would be helpful to include it in medical insurance when the therapies are mature, so that the general public can afford it, and physicians can have more patients in their medical records to justify and improve pre-clinical research.

Above all, the strength of our review is the inclusion and combination of recent studies in this area systematically. Most previous reviews on stem cell therapies for AD had focused solely on: (1) pathology [110] (pre-clinical only) or (2) a specific type of stem cell (e.g., MSC only [111]); previous comprehensive reviews also comprised many articles that were not recent (i.e., within 5 years) [112]. Our scoping review includes recent studies on a variety of stem cells, and categorizes their pathological impacts on treating AD. However, the weakness of our review is the inability to include statistical analysis and quantitative comparisons. This is because the statistical methods in the included studies vary extensively, and it is difficult to standardize the results with specific criteria. Hopefully with more clinical studies emerging in the future there could be the potential to compare the results of studies by the standard of patient survival rate.

## 5. Conclusions

In conclusion, stem cell therapy is a promising treatment for Alzheimer’s disease (AD), but it is still in the process of development. There are four major types of stem cells for AD therapy: neural stem cells, mesenchymal stem cells, embryonic stem cells, and induced pluripotent stem cells. All of them have applications in the studies of (1) cellular level, (2) animal model, and (3) clinical level of AD. In general, there were many more types of stem cells studied on the cellular level and in an animal model, than on the clinical level. We suggest future studies to increase research on various types of stem cells, and include cross-disciplinary research with other diseases. In the future, there could also be improvements in the timeliness of research and individualization for stem cell therapies for AD.

## Figures and Tables

**Figure 1 biomedicines-11-00120-f001:**
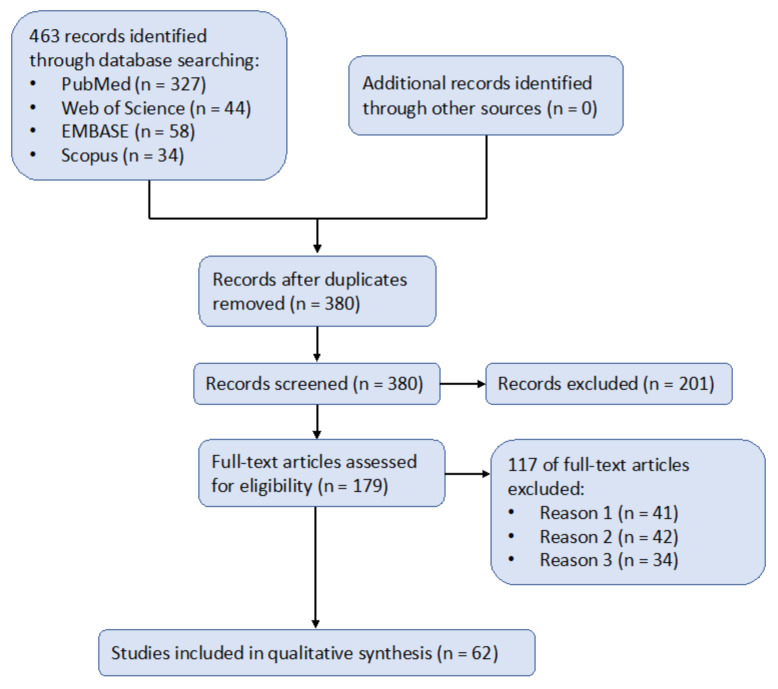
Flow chart of the literature search following PRISMA 2020 guidelines. Searching databases were used as detailed in the main text. The reasons for exclusion of articles were as follows. Reason 1: stem cell was mentioned not as a direct therapeutic method, but as a medium for other medical treatments or preclinical research. Reason 2: the article included brief mentioning of Alzheimer’s disease, but was not targeted toward Alzheimer’s disease. Reason 3: Ineligible exposure group.

**Figure 2 biomedicines-11-00120-f002:**
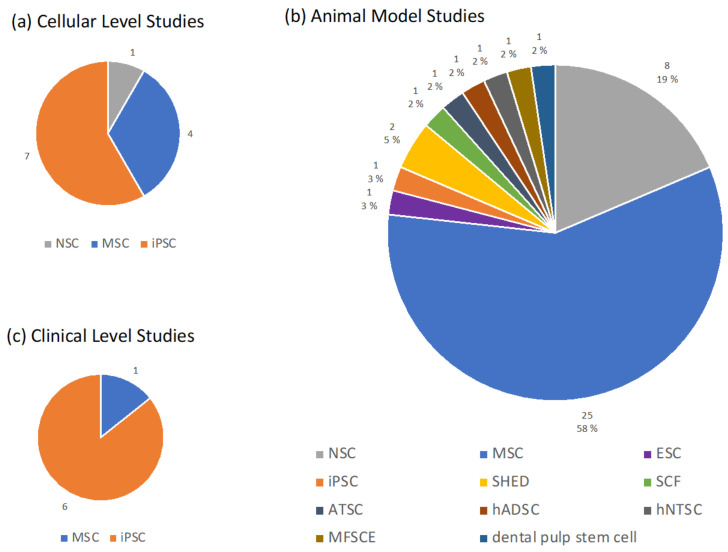
Distribution of stem cell types for AD therapies. (**a**) Cellular level studies, (**b**) Animal model studies, (**c**) Clinical level studies. Abbreviations: MSC (mesenchymal stem cell), NSC (neural stem cell), iPSC (induced pluripotent stem cell), ESC (embryonic stem cell), SHED (stem cell from human exfoliated deciduous teeth), SCF (stem cell factor), ATSC (adipose tissue mesenchymal stem cell), hADSC (human adipose-derived stem cell), hNTSC (human neural crest-derived nasal turbinate stem cell), MFSCE (membrane-free stem cell).

**Figure 3 biomedicines-11-00120-f003:**
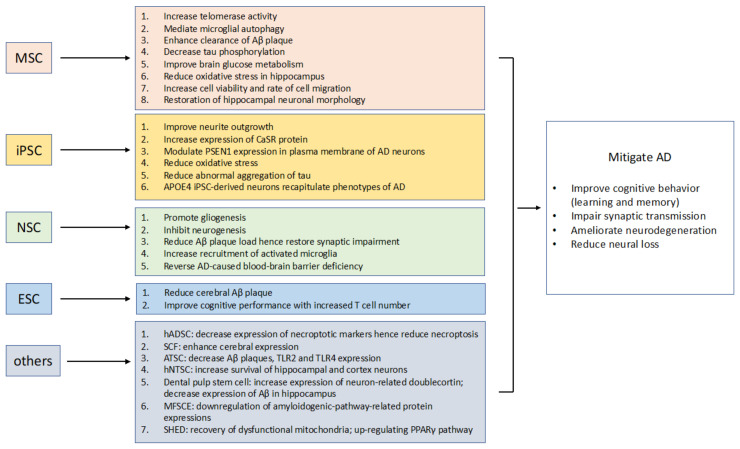
Collection of pathological effects for different types of stem cells. Abbreviations: MSC (mesenchymal stem cell), NSC (neural stem cell), iPSC (induced pluripotent stem cell), ESC (embryonic stem cell), SHED (stem cell from human exfoliated deciduous teeth), SCF (stem cell factor), ATSC (adipose tissue mesenchymal stem cell), hADSC (human adipose-derived stem cell), hNTSC (human neural crest-derived nasal turbinate stem cell), MFSCE (membrane-free stem cell).

**Table 1 biomedicines-11-00120-t001:** Summary of cellular level studies on stem cell therapy for AD (2017–2022).

	Author	Year	Type of Stem Cell	Intervention	Control Group	Measured Outcomes of Interest	Reference
Animal-induced stem cells	Farahzadi et al.	2020	BM-MSC	Injection of BM-MSCs with Aβ-treated neural cells	Aβ-treated neural cells without BM-MSC injection	Significant increase in telomerase activity, which indicates mitigation of AD	[48]
Human-oriented stem cells	Xu et al.	2018	ucMSC	Culture ucMSCs to BV2 cell line with Aβ25–35 oligomers	BV2 cell line without culturing ucMSCs	MSCs inhibited proliferation of BV2 cells, mediated microglial autophagy, and enhanced clearance of Aβ, hence mitigating AD	[49]
	Lin et al.	2018	iPSC	Using CRISPR/Cas9 to generate isogenic iPSCs harboring homozygous APOE4 alleles from unaffected parental APOE3 cells	Parental APOE3 cells without injection of isogenic iPSC lines	APOE4 iPSC-derived neurons recapitulate phenotypes of AD at multiple levels, including increased synapse number and elevated Aβ_42_ secretion	[46]
	Chang et al.	2019	iPSC	Injection of a potential Aβ aggregation reducer indole compound NC009–1 with iPSC on live human neurons from AD patients	AD neurons without injection of iPSC	Improved neurite outgrowth in AD-iPSC-derived neurons	[50]
	Coronel et al.	2019	hNSC	Transiently increase APP level in hNSCs	hNSCs without increasing APP level	Promoted gliogenesis and inhibited neurogenesis in hNSCs	[51]
	Giudice et al.	2019	iPSC	Injection of γ-secretase inhibitor to iPSC-derived neurons from AD patients	AD neurons without injection of iPSC-derived neurons	Increased expression of CaSR protein and modulated PSEN1 expression in plasma membrane of AD neurons	[52]
	Marzano et al.	2019	hiPSC	Injection of hiPSC-derived extracellular vesicles (EVs) to AD-associated SY-UBH cell lines	SY-UBH cells without hiPSC injection	hiPSC-derived EVs exhibited neural protective abilities in Aβ_42_ oligomer-treated cultures, enhancing cell viability and reducing oxidative stress	[53]
	Meyer et al.	2019	iPSC	iPSCs generated from sporadic AD (SAD) patients	iPSCs generated from age-matched controls of familial AD (FAD) patients	SAD iPSC lines showed a significant level of increase in neural differentiation-related gene expression, leading to premature neuronal differentiation, and reduced neural progenitor cell renewal	[54]
	van der Kant et al.	2019	iPSC	iPSC-derived neurons from SAD patients	iPSC-derived neurons from FAD patients	Allosteric activation of CYP46A1 lowered cholesteryl esters (CE) in iPSC-derived neurons, indicating a druggable CYP46A1-CE-Tau axis in AD.	[55]
	Zhang et al.	2020	hucMSC	Treatment of hucMSC conditioned medium to AD cellular model established by okadaic acid-treated SH-SY5Y cells	SH-SY5Y cells without treatment of hucMSC	hucMSCs significantly decreased tau phosphorylation at Thur181 level, and alleviated intracellular and mitochondrial oxidative stress in AD cells	[56]
	Zhao et al.	2020	iPSC	Cerebral organoid model using iPSCs with APOE3 or APOE4 genotype from individuals with AD dementia	Cerebral organoid model using iPSCs with APOE3 or APOE4 genotype from individuals with normal cognition	Cerebral organoids model using iPSCs from AD patients carrying APOE4 show greater apoptosis and decreased synaptic integrity	[47]
	Chen et al.	2021	MSC	Culture MSC-exosomes into human neural cell with AD mutations	Human neural cells without culturing MSC-exosomes	Improvement in brain glucose metabolism and cognitive functioning	[57]

**Table 2 biomedicines-11-00120-t002:** Summary of animal model studies on stem cell therapy for AD in chronological order (2017–2022).

Author	Year	Type of Stem Cell	Intervention	Control Group	Measured Outcomes of Interest	Reference
Boutajangout et al.	2017	hucMSC	Direct injection of hucMSCs into carotid artery of APP/PS1 Tg AD model mice	AD mice without injection of hucMSCs	Reduction of amyloid beta burden in cortex and hippocampus which correlated with a reduction of cognitive loss	[58]
Cui et al.	2017	ucMSC	Intravenous transplantation of ucMSCs to Tg2576 mice, which express AD-like APP pathological forms	Tg2576 mice without transplantation of ucMSCs	Reduced oxidative stress in hippocampus of AD mice due to decrease of malondialdehyde (MDA) and hence up-regulated neuronal synaptic plasticity	[59]
Harach et al.	2017	MSC	Administer MSC intravenously to APP/PS1 transgenic mice that developed cerebral Aβ	APPPS1 transgenic mice without administering MSCs	Reduced soluble cerebral Aβ levels and increased Aβ-degrading enzymes to modulate cerebral cytokines	[60]
Kim et al.	2017	hUCB-MSC	Injection of hUCB-MSCs in C3H/C57 wild-type mice	C3H/C57 mice without injection of hUCB-MSC	Increased cell viability and rate of cell migration in the brain intracerebroventricular route, hence mitigating AD.	[61]
Han et al.	2018	BM-MSC	Exposure to MicroRNA let-7f-5p modified MSCs in vitro in C57BL/6 mice	C57BL/6 mice without MSC transplantation	Increased caspase-3 expression and hence decreased cytotoxicity for AD models	[62]
McGinley et al.	2018	hNSC	Transplantation of hNSC targeted to fimbria fornix of APP/PS1 murine	APP/PS1 mice without transplantation of hNSCs	Reduced amyloid plaque load and increased recruitment of activated microglia, indicating functional and pathological improvements in AD mice	[23]
Oh et al.	2018	sRAGE-MSC	Injection of sRAGE-MSCs with Aβ_1–42_ to entorhinal cortices of male Sprague Dawley rats	Sprague Daley rats without injection of sRAGE-MSCs	Longer survival time for mice with sRAGE-MSC injection due to prevented apoptosis, and decrease in neuron numbers	[63]
Wang et al.	2018	MSC	Intracerebroventricular injection of MSC-derived extracellular vesicles to APP/PS1 mice	APP/PS1 mice without injection of MSCs	Alleviated exogenous Aβ-induced iNOS mRNA and protein expression, therefore improved cognitive behavior and rescued impairment of CA1 synaptic transition and long-term potentiation in AD mice	[64]
Wang et al.	2018	ucMSC	Injection of ucMSCs with resveratrol to hippocampus of AD mice	AD mice without ucMSC injection	Improved learning and memory, enhanced neurogenesis, and alleviated neural apoptosis in AD mice	[65]
Wei et al.	2018	BM-MSC	Injection of BM-MSCs on APP/PS1 mice	APP/PS1 mice without BM-MSC injection	Reduced Aβ_1–42_ content and BACE1 gene expression; ameliorated symptoms of AD	[66]
Yu et al.	2018	BM-MSC	Transplantation of BM-MSCs to murine models with AD	AD murine model without BM-MSC transplantation	Increased protein expression of seladin-1 and nestin, and hence reduced neurodegeneration	[67]
Esmaeilzade et al.	2019	MSC	Incubation of MSCs with dimethyloxalylglycine (DMOG) in Aβ-injected rats	Aβ-injected rats without MSC incubation	Increased cell viability, migration, and expression of CXCR4, CCR2, Nrf2, and HIF-1α, which enhanced antioxidant capacity in the hippocampus	[68]
Hu et al.	2019	ucMSC	Transplantation of ucMSCs with brain-derived neurotrophic factor (BDNF) to mice with Aβ_1–42_	AD mice without transplantation of ucMSCs	Significantly improved spatial learning and memory abilities in the AD rats; increased release of acetylcholine and ChAT expression in the hippocampus	[69]
Nasiri et al.	2019	AD-MSC	Injection of melatonin-pretreated AD-MSCs to male Wistar rats	Male Wistar rats without injection of AD-MSCs	Cleared Aβ deposition and reduced microglial cells	[70]
Reza-Zaldivar et al.	2019	MSC	Administration of MSC-derived exosomes to amyloid 1–42 treated AD mice	AD mice without injection of MSC-derived exosomes	Alleviated beta amyloid 1–42-induced cognitive impairment	[27]
Eftekharzadeh et al.	2020	hADSC	Intravenous injection of hADSCs to murine model of AD	Murine model without administration of hADSCs	hADSCs significantly decreased the expression of necroptotic markers and reduced necroptosis and declined death of neuronal cells in the hippocampus of AD rats	[71]
Guo et al.	2020	SCF	Subcutaneously injection of SCF to APP/PS1 transgenic mice with C57BL/6J genetic background	APP/PS1 transgenic mice without injection of SCF	Increased association of TREM2^+^/Iba1^+^ cells with Aβ plaques and enhanced cerebral expression that ameliorated AD pathology at late stage	[72]
Kim et al.	2020	ES-MSC	Intra-arterial administration of ES-MSCs in AD rat model	AD rats without administration of ES-MSCs	Significantly inhibited Aβ-induced cell death in hippocampus and promoted autophagolysosomal clearance of Aβ	[33]
Liu et al.	2020	NSC	Injection of NSC-derived exosomes to 5 × FAD mice	5 × FAD mice without injection of NSC-derived exosomes	Reversed AD-caused blood–brain barrier deficiency	[73]
Mehrabadi et al.	2020	ATSCs	Injection of hypoxic-conditioned medium from ATSCs intravenously to AD mice	AD mice without injection of ATSCs	Decreased beta amyloid plaques, TLR2 and TLR4 expression and enhanced neuronal survival	[74]
Park et al.	2020	BM-MSC	Intravenous injection of BM-MSCs to 3xTg AD mice	AD mice without injection of BM-MSCs	Enhanced memory function and less β-amyloid-immunopositive plaques	[75]
Park et al.	2020	NSC	Transplantation of NSCs into APPswe/PS1dE9 AD model mice	AD mice without NSC transplantation	Increased ACh (acetylcholinesterase) level and improved learning and memory function	[76]
Ramezani et al.	2020	BM-MSC	Transplantation of BM-MSCs to male Wistar rats	Wistar rats without BM-MSC transplantation	Enhanced learning, cognition and memory that mitigated neurodegeneration of AD	[77]
Zhao et al.	2020	ESC	Transplantation of mouse ESC-derived thymic epithelial progenitors into AD mice	Without ESC transplantation	Reduced cerebral Aβ plaque load and improved cognitive performance with increased T cell number	[78]
Zhu et al.	2020	NSC	Implantation of NSCs to hippocampus of APP/PS1 Tg (transgenic) AD mice	AD mice without NSC implantation	Protected cholinergic neurons, restored synaptic impairment in amyloid precursor and hence improved learning and memory function	[79]
Apodaca et al.	2021	hNSC	Intravenous injection of hNSC-derived extracellular vesicles on 5xFAD accelerated transgenic AD mice	AD mice without injection of hNSC-derived extracellular vesicles	Reduced dense core Aβ plaque accumulation and microglial activation, restoration of homeostatic levels of circulating pro-inflammatory cytokines in AD mice, with improved cognition and synaptic function	[25]
Armijo et al.	2021	iPSC	Stereotaxically injection of iPSC-derived neural precursors to hippocampus of 3xTg AD mice	AD mice without injection of iPSC-derived neural precursors	Improved memory, synaptic plasticity, and reduced brain pathology, including a reduction of amyloid and tangles deposits	[37]
Cone et al.	2021	MSC	Intranasally injection of NSC derived extracellular vesicles to non-transgenic 5xFAD mice	AD mice without NSC injection	Lowered Aβ plaque load in the hippocampus. Less colocalization between GFAP and Aβ plaques and hence better cognition functions	[80]
Huang et al.	2021	Nanoformulation-mediated NSC	Injection of nanoformulation-mediated NSC into APPswe/PS1dE9 double transgenic mouse model	Mouse model without NSC injection	Improved neural regeneration, and efficient and long-lasting Aβ degradation	[81]
Jeong et al.	2021	hucMSC	Intravenous transplantation of hucMSCs into Aβ injected AD animal model	AD animal model without hucMSC transplantation	Superior neurogenesis and anti-inflammation properties with increased NEP in hippocampus	[82]
Kuo et al.	2021	MSC	Intracerebroventricular administration of MSC-conditioned medium to Aβ-induced rat model	Rat model without administration of MSC	Decreased expression of tight junction proteins, SIRT1 and β-catenin, which attenuated retinal pathology of AD	[83]
Lim et al.	2021	hNTSC	Transplantation of hNTSC into 5xFAD transgenic AD mice	AD mice without hNTSC transplantation	Reduced Aβ_42_ levels and plaque formation in the brain, increased survival of hippocampal and cortex neurons	[84]
Lu et al.	2021	hNSC	Intranasal transplantation of hNSCs into APP/PS1 transgenic AD mice	AD mice without hNSC transplantation	Attenuated beta-amyloid accumulation by upregulating the expression of beta-amyloid degrading enzymes, insulin degrading enzymes and neprilysin, which ameliorated neuroinflammation, cholingergic dysfunction, and synaptic loss	[85]
Neves et al.	2021	BM-MSC	Administer of allogeneic BM-MSCs intravenously in 3xTg AD mice	AD mice without administering BM-MSCs	Reduced β-secretase cleavage of amyloid precursor protein and decreased tau phosphorylation	[86]
Santamaria et al.	2021	MSC	In vivo systematic administration of MSCs to APP/PS1 AD mice	AD mice without administration of MSCs	Induced persistent memory recovery and reduced plaques with β-amyloid oligomers	[87]
Wang et al.	2021	MSC	Tail-vein injection of MSC-derived small extracellular vesicles into APP/PS1 AD mice	AD mice without injection of MSCs	Restored hippocampal neuronal morphology, with improved cognitive impairments and reduced neuronal loss	[88]
Zhang et al.	2021	Dental pulp stem cell	Injection of 5 × 10 dental pulp stem cells into the hippocampus of AD mice	AD mice without injection of dental pulp stem cells	Increased expression of neuron-related doublecortin, NeuN, and neurofilament 200 in the hippocampus with decreased expression of Aβ, hence improving cognitive and behavioral abilities	[89]
Choi et al.	2022	MFSCE	Treatment of MFSCE on Aβ_25–35-_injected AD mice	AD mice without MFSCE treatment	Suppressed Bax and cleaved caspase-3 protein expression, downregulated amyloidogenic-pathway-related protein expressions and hence improved cognitive functioning	[90]
Guo et al.	2022	SHED	SHEDs cultured in vitro and injected into AD SAMP8 mice by caudal vein	AD mice without SHED injection	Improved cognitive ability and reversed memory loss through the recovery of dysfunctional mitochondria	[91]
Liu et al.	2022	BM-MSC	Lateral ventricle administration of BM-MSCs to adult C57BL/6 AD mice	AD mice without administration of BM-MSCs	Inhibited hyper activation of microglia and astrocytes in the hippocampus of AD mice	[92]
Wang et al.	2022	hucMSC	Injection of hucMSCs with Fe_3_O_4_ polydopamine nanoparticles into APP/PS1 transgenic mice	AD mice without hucMSC injection	Improved memory and cognitive ability of AD by increased expression of brain-derived neurotrophic factor	[93]
Zhang et al.	2022	NSC	Transplantation of NSCs into hippocampal CA1 region of rTg (tau P301L) 4510 mouse model	Mouse model without transplantation of NSCs	Reduced abnormal aggregation of tau, and hence improvements in short-term memory	[94]
Zhang et al.	2022	SHED	Injection of SHED into SAMP8 AD mice	AD mice without SHED injection	Relieved AD symptoms by up-regulating PPARγ pathway	[95]

**Table 3 biomedicines-11-00120-t003:** Summary of clinical level studies on stem cell therapy for AD in chronological order (2017–2022).

Author	Year	Type of Stem Cell	Intervention	Control Group	Measured Outcomes of Interest	Reference
Wang et al.	2018	iPSC	Induction of iPSC with Sendai-virus delivery system	N/A	Obtain iPSC cell line (ZZUi009-A) from AD patient with PSEN1 gene mutation	[96]
Wang et al.	2019	iPSC	Induction of iPSC with novel MEOX2 mutation in a family with AD	N/A	Obtain iPSC cell line (ZZUi0013-A) with episomal plasmids expressing OCT3/4, SOX2, KLF4, LIN28, and L-MYC genes	[97]
Dai et al.	2020	iPSC	Generation of iPSC from peripheral blood mononuclear cells with AD patient with APOE3/4 genotype	N/A	With immunocytochemistry iPSC displayed potential to differentiate spontaneously into three germ layers in vitro	[98]
Kim et al.	2021	hUCB-MSC	Intracerebroventricular injection of hUCB-MSCs to AD patients	AD patients without hUCB-MSC injection	Phase I clinical trial showed significant effect in mitigating neurodegeneration despite few adverse events (fever, headache, vomit)	[30]
Wang et al.	2021	iPSC	Induction of iPSC from male AD patient with APOE-ε4/ε4 alleles	N/A	Obtain iPSCs with differentiation potential for treating neurological disorder and multiple sclerosis	[99]
Wang et al.	2021	iPSC	Induction of iPSC with APP gene mutation in a female patient with AD	N/A	Obtain iPSC cell line (ZZUi0024-A) with dermal fibroblasts expressing OCT3/4, SOX2, KLF4, LIN28, and L-MYC genes	[100]
Lee et al.	2022	iPSC	Induction of iPSC from AD patients with mtDNA mutations	N/A	mtDNA mutations induced growth advantage with higher viability and proliferation, lower mitochondrial respiration, and membrane potential	[101]

## Data Availability

No new data were created or analyzed in this study. Data sharing does not apply to this article.

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
