# Peer review of "Stem Cell Therapy for Alzheimer’s Disease: A Scoping Review for 2017–2022"

_biomedicines, 2023, doi:10.3390/biomedicines11010120_

Round 1
Reviewer 1 Report
This review includes a promising treatments for Alzheimer´s disease, including recent studies on a variety of 376 stem cells and categorizes their pathological impacts on treating this neurodegenerative disorder.
Author Response
Dear Reviewer 1,
Thank you so much for your encouraging comments!
We sincerely appreciate your acknowledgment of our manuscript.
Best regards.
Reviewer 2 Report
This review aims to represent the importance of “Stem cell therapy” in “Alzheimer’s disease”. At first, the authors tried to introduce available therapeutic strategies for Alzheimer’s disease. After that, they represent some “types of stem cells” (1) Neural stem cells, 2) Mesenchymal stem cells, 3) Embryonic stem cells, and 4) induced pluripotent stem cells for “Alzheimer’s disease” therapy. however the topic is an interesting one and MS is well organized but some points should be considered.
In details, in the abstract and introduction, the authors failed to define the gap in the field and to bring the rational why this study was needed. They did not write about the current therapeutic options for Alzheimer’s disease (Trophic factors, Insulin therapy, Low-level laser therapy, signaling pathways, herbal medicine, mitochondrial calcium targeting, healthy lifestyle, etc.).
Furthermore, they could not introduce the problems/complications of “stem cell therapy” such as 1) Isolation and enough expansion of neural stem cells from the CNS are so tough in vitro and require culture medium supplemented with mitogenic growth factors, such as epidermal growth factor and basic fibroblast growth factor. Dramatically, the application of neural stem cells to large-scale clinical states is lacking. 2) The risk of tumorigenicity and infection of the induced pluripotent stem cells‐ derived donor cells associated with induced pluripotent stem cells- based therapy. 3) The side effects of tumor formation and graft failure for embryonic stem cells. 4) Some organelle dysfunction in mesenchymal stem cell therapy. Accordingly, the authors should represent more knowledge/examples to show the advantages/disadvantages of stem cell therapy (ethical issues, technical limitations, etc.) in clinics.
The paper needs proper punctuation and minor language editing. Some sentences are unclear.
Also, the authors can design and prepare attractive figures for the paper.
Author Response
Dear Reviewer 2,
Thank you so much for giving us the opportunity to revise our manuscript entitled “Stem Cell Therapy for Alzheimer’s Disease: A Scoping Review for 2017-2022”. (manuscript ID: biomedicines-2029118). We highly appreciate your constructive comments and suggestions to our manuscript. We have carefully studied your comments and made revisions with “Track Changes” function of Word document in the revised manuscript, which we would like to submit for your kind consideration. As attached, please find the detailed reply to your comments.
“In details, in the abstract and introduction, the authors failed to define the gap in the field and to bring the rational why this study was needed. They did not write about the current therapeutic options for Alzheimer’s disease (Trophic factors, Insulin therapy, Low-level laser therapy, signaling pathways, herbal medicine, mitochondrial calcium targeting, healthy lifestyle, etc.).”
Indeed, according to your suggestion, we have added current therapeutic options for Alzheimer’s disease and their limitations in the introduction section (Section 1.1 Alzheimer’s Disease and Its Treatments, Page 1-2, line 40-69) under the paragraph on drugs for treating Alzheimer’s disease. We addressed all of the therapeutic options mentioned in your comments, including: trophic factors, insulin therapy, low-level laser therapy, signaling pathways, herbal medicine, mitochondrial calcium targeting, and healthy lifestyle. Because none of the therapeutic methods show a definite ability to cure Alzheimer’s disease, stem cell therapy, with its promising therapeutic potential, should be investigated. Therefore, this scoping review is needed to map the most recent development of stem cell therapy for Alzheimer’s disease to help future researchers more conveniently refer to the development and mechanism of stem cell therapy for Alzheimer’s disease.
“Furthermore, they could not introduce the problems/complications of “stem cell therapy” such as 1) Isolation and enough expansion of neural stem cells from the CNS are so tough in vitro and require culture medium supplemented with mitogenic growth factors, such as epidermal growth factor and basic fibroblast growth factor. Dramatically, the application of neural stem cells to large-scale clinical states is lacking. 2) The risk of tumorigenicity and infection of the induced pluripotent stem cells‐ derived donor cells associated with induced pluripotent stem cells- based therapy. 3) The side effects of tumor formation and graft failure for embryonic stem cells. 4) Some organelle dysfunction in mesenchymal stem cell therapy. Accordingly, the authors should represent more knowledge/examples to show the advantages/disadvantages of stem cell therapy (ethical issues, technical limitations, etc.) in clinics.”
Thank you for this constructive comment. We have included all your suggestions for the potential problems and complications of stem cell therapy in the discussion section (Section 4. Discussion, Page 13-14, line 426-441). We have addressed all 4 points that you mentioned in your comments and put citations for each of them accordingly. We also specified ethical issues and technical limitations as you pointed out in 4).
“The paper needs proper punctuation and minor language editing. Some sentences are unclear.”
Thank you for your comment. We have revised the punctuation and language for the manuscript and made sure that all sentences are delivered clearly.
“Also, the authors can design and prepare attractive figures for the paper.”
Thank you for your suggestion. We have colored Figure 3 (Page 7) according to different types of stem cells. We also revised Figure 1 (Page 5) with coloring for each box and increased its resolution for the convenience of viewing. Hopefully in this way, we can help prepare the figures to be more attractive to our audiences and also make them more comprehensible.
Above all, we hope that the revision and correction will meet your approval.
Best regards.

Reviewer 3 Report
There are also many papers on the establishment of iPSCs from actual patients for the elucidation of the pathogenesis of Alzheimer's disease and the development of treatment methods. If the title of this review contains the word “therapy,” I think that such papers should be distinguished.
Author Response
Dear Reviewer 3,
Thank you so much for giving us the opportunity to revise our manuscript entitled “Stem Cell Therapy for Alzheimer’s Disease: A Scoping Review for 2017-2022”. (manuscript ID: biomedicines-2029118). We highly appreciate your constructive comments and suggestions to our manuscript. We have carefully studied your comments and made revisions with “Track Changes” function of Word document in the revised manuscript, which we would like to submit for your kind consideration. As attached, please find the detailed reply to your comments.
“There are also many papers on the establishment of iPSCs from actual patients for the elucidation of the pathogenesis of Alzheimer's disease and the development of treatment methods. If the title of this review contains the word “therapy,” I think that such papers should be distinguished.”
Thank you for your constructive suggestion. There are indeed many papers using iPSCs to elucidate the pathogenesis of Alzheimer’s disease. In the clinical-level studies that we included in our initial manuscript, 6 out of 7 included studies are based on iPSCs from actual patients (Section 3.3 Clinical Level, Page 12). Due to the concern of timeliness, and the fact that there are already many reviews on stem cell therapy for Alzheimer’s disease in the past, we only included studies from 2017 to 2022 (the most recent 5 years), which is why the number of studies appears to be smaller than your suggestion.
In addition, we were hesitating at first to include some of the studies using iPSCs only as medium for other medical treatments or preclinical research. (Section 2.2 Inclusion and Exclusion Criteria, Reason 1; Page 6, line 269-274) But after viewing your comment, we re-considered the studies that we excluded for this reason and found that there are 4 studies that qualify for inclusion based on your suggestion (Lin et al., Meyer et al., van der Kant et al., Zhao et al.). These studies are included in Section 3.1 (Cellular level, Page 8-9, Table 1) rather than clinical level because in these studies iPSCs were drawn from patients and experimented in a laboratory setting. In particular, among the 4 newly included studies, 2 shown the relationship between APOE4 genotype with Alzheimer’s disease. We described this finding in the explanations above Table 1 (Page 7, line 329-331). We have also adjusted the proportion of iPSC studies in Figure 2 (Page 6) and Figure 3 (Page 7), as well as the number of included studies in Figure 1 (Page 5).
Above all, we hope that the revision and correction will meet your approval.
Best regards.

Reviewer 4 Report
Summary: The manuscript entitled “Stem Cell Therapy for Alzheimer’s Disease: A Scoping Review for 2017-2022” offers a concise review of the therapeutic aspect of stem cell therapy against Alzheimer’s disease published since 2017. The authors comprised recent publications on the impact of different stem cells on AD. After a thorough review, I recommend publishing the manuscript after a minor revision.
1. The manuscript is organized, but it is suggested to check it one more time, as there are some typo errors.
2. Why the authors selected the time frame of 2017-2022? Any specific reason?
3. The authors mentioned, “One of the reasons is targeting the wrong pathological substrates” later in the manuscript, different references were cited where different Alzheimer models were used, including APP/PS1, 5xFAD with positive results after stem cell therapy. The underlying mechanism was amyloid-β plaque accumulation. There are numerous reports available where amyloid-β plaque is the target substrate. The discussion states, “The reason behind massive pre-clinical studies is the unclear pathology for AD with stem cell therapy despite its potential effectiveness”. It means amyloid -β plaque is one of the pathologies in AD progression. Therefore, instead of the “wrong pathological substrate”, “unclear pathology” should be the preferred term to address the issue.
Author Response
Dear Reviewer 4,
Thank you so much for giving us the opportunity to revise our manuscript entitled “Stem Cell Therapy for Alzheimer’s Disease: A Scoping Review for 2017-2022”. (manuscript ID: biomedicines-2029118). We highly appreciate your constructive comments and suggestions for our manuscript. We have carefully studied your comments and made revisions with “Track Changes” function of Word document in the revised manuscript, which we would like to submit for your kind consideration. As attached, please find the detailed reply to your comments.
“1. The manuscript is organized, but it is suggested to check it one more time, as there are some typo errors.”
Thank you for your comment. We have checked the language of our manuscript and revised typo errors in the contents.
“2. Why the authors selected the time frame of 2017-2022? Any specific reason?”
The reason why we selected the time frame of 2017 to 2022 is primarily that we wish to achieve the timeliness of our scoping review. There is already an abundance of previous review articles that comprehensively summarized the development of stem cell therapy for Alzheimer’s disease in the past. However, the most recent 5 years (i.e. 2017-2022) still lack a scoping review. Among currently published review articles, Duncan et al. (“Alzheimer’s Disease, dementia, and stem cell therapy”) published in 2017 is the most recent and comprehensive review of stem cell therapy for Alzheimer’s disease (we included this article as reference 16, detailed reference information is in the Reference section, Page 16, line 546). Many other review articles are either minireviews that lack comprehensiveness or reviews that focus only on a specific type of stem cell therapy.
In Section 2.1 (Search Strategy, Page 5, line 238-240), we addressed the reason why we selected published articles in the range of years 2017 to 2022: “in order to evaluate stem cell therapy’s effectiveness on Alzheimer’s disease up to date”. By selecting the most recently published articles, we hope to help researchers in this field find frontier studies they need more conveniently.
In the last paragraph of Section 4 (Discussion, Page 14, line 465-469), we also explained that: “Most previous reviews on stem cell therapies for Alzheimer’s disease had focused solely on: (1) pathology (pre-clinical only) or (2) a specific type of stem cell (e.g. MSC only); previous comprehensive reviews also comprised many articles that were not recent (i.e. within 5 years)”. However, almost all of the references in our scoping review are within the past 5 years to make sure that we can provide the most recent information to our readers.
“3. The authors mentioned, “One of the reasons is targeting the wrong pathological substrates” later in the manuscript, different references were cited where different Alzheimer models were used, including APP/PS1, 5xFAD with positive results after stem cell therapy. The underlying mechanism was amyloid-β plaque accumulation. There are numerous reports available where amyloid-β plaque is the target substrate. The discussion states, “The reason behind massive pre-clinical studies is the unclear pathology for AD with stem cell therapy despite its potential effectiveness”. It means amyloid -β plaque is one of the pathologies in AD progression. Therefore, instead of the “wrong pathological substrate”, “unclear pathology” should be the preferred term to address the issue.”
Thank you so much for your detailed suggestion. We have changed “wrong pathological substrate” to “unclear pathology” in Section 1.1 (Alzheimer’s Disease and Its Treatments, Page 1, line 31-32). We really appreciate your explanation with professional expertise.
Above all, we hope that the revision and correction will meet your approval.
Best regards.

Round 2
Reviewer 2 Report
The authors adequately addressed my concerns, and added information needed toimproved the understanding and the quality of the manuscript.
Therefore, I recommend publication in the journal.
Author Response
Dear Reviewer 2,
Thank you so much for your encouraging comments!
We sincerely appreciate your acknowledgment of our manuscript.
Best regards.